# Modulation of Nitric Oxide Synthases by Oxidized LDLs: Role in Vascular Inflammation and Atherosclerosis Development

**DOI:** 10.3390/ijms20133294

**Published:** 2019-07-04

**Authors:** Micaela Gliozzi, Miriam Scicchitano, Francesca Bosco, Vincenzo Musolino, Cristina Carresi, Federica Scarano, Jessica Maiuolo, Saverio Nucera, Alessia Maretta, Sara Paone, Rocco Mollace, Stefano Ruga, Maria Caterina Zito, Roberta Macrì, Francesca Oppedisano, Ernesto Palma, Daniela Salvemini, Carolina Muscoli, Vincenzo Mollace

**Affiliations:** 1Institute of Research for Food Safety & Health IRC-FSH, University Magna Graecia, 88021 Catanzaro, Italy; 2Nutramed S.c.a.r.l. Complesso Ninì Barbieri, Roccelletta di Borgia, 88021 Catanzaro, Italy; 3Department of Medicine, Chair of Cardiology, University of Rome Tor Vergata, 00133 Rome, Italy; 4Department of Pharmacology and Physiology, Saint Louis University School of Medicine, St. Louis, MO 63104, USA; 5IRCCS San Raffaele Pisana, 00163 Roma, Italy

**Keywords:** oxidized LDLs, constitutive NO synthase cNOS, inducible NO synthase (iNOS), endothelial dysfunction

## Abstract

The maintenance of physiological levels of nitric oxide (NO) produced by eNOS represents a key element for vascular endothelial homeostasis. On the other hand, NO overproduction, due to the activation of iNOS under different stress conditions, leads to endothelial dysfunction and, in the late stages, to the development of atherosclerosis. Oxidized LDLs (oxLDLs) represent the major candidates to trigger biomolecular processes accompanying endothelial dysfunction and vascular inflammation leading to atherosclerosis, though the pathophysiological mechanism still remains to be elucidated. Here, we summarize recent evidence suggesting that oxLDLs produce significant impairment in the modulation of the eNOS/iNOS machinery, downregulating eNOS via the HMGB1-TLR4-Caveolin-1 pathway. On the other hand, increased oxLDLs lead to sustained activation of the scavenger receptor LOX-1 and, subsequently, to NFkB activation, which, in turn, increases iNOS, leading to EC oxidative stress. Finally, these events are associated with reduced protective autophagic response and accelerated apoptotic EC death, which activates atherosclerotic development. Taken together, this information sheds new light on the pathophysiological mechanisms of oxLDL-related impairment of EC functionality and opens new perspectives in atherothrombosis prevention.

## 1. Introduction

Endothelial cells (ECs) are the major component of the innermost layer of mammal blood vessels, thereby representing a natural barrier, which functionally serves to maintain the bloodstream separate from extra-vascular tissues. As a consequence, the integrity of the vascular endothelium represents a key pre-requisite for regulating regional blood flow and many other mechanisms that contribute in modulating vascular responses [1,2], including the inhibition of atherothrombotic disease development [3].

Among pathophysiological factors that have been identified to generate endothelial dysfunction, overproduction of oxidized low-density lipoproteins (oxLDLs) appears to play a crucial role in the development of atherosclerosis. Indeed, evidence has been accumulated demonstrating that oxLDLs produce pathophysiological effects, characterized by the release of proinflammatory cytokines, overexpression of cell adhesion molecules, monocyte chemotactic protein-1, and smooth muscle cell (SMC) growth factors, and impairment of endothelium-dependent vasorelaxation [4,5]. In addition, EC dysfunction due to the overproduction of oxLDLs leads to imbalanced activation of nitric oxide synthase (NOS,) expressed constitutively by ECs (eNOS), thereby facilitating the activation of the inducible isoform of this enzyme (iNOS), which, in turn, enhances inflammatory processes within the vascular wall and contributes to atherosclerosis progression [6,7,8]. However, despite clear evidence suggesting that an imbalanced modulation of eNOS/iNOS derives from high oxLDL production in hyperlipemic patients and that this process underlies some relevant aspects of atherosclerosis progression, the mechanisms through which oxLDLs impair endothelial function, thus promoting atherosclerosis, remain unclear [9,10].

Consistent data show that the basal release of NO and the subsequent endothelial-dependent vasodilation are impaired during the development of hypercholesterolemia [11,12,13,14]. In particular, it has been assessed that local NO release is attenuated in patients undergoing serum levels of LDL up to 160 mg/dL [15,16]. This has been correlated with many pathophysiological mechanisms. Among them, the upregulation of the enzyme arginase causes a reduced availability of L-arginine, which represents the natural eNOS substrate [11,12]. Moreover, a reduction in the eNOS cofactor tetrahydrobiopterin (BH4) occurs [11]. On the other hand, the potential role of oxLDLs in triggering molecular pathways, leading to vascular inflammation, via the up-regulation of its receptor LOX-1 and the subsequent NO dysregulation, need to be explored.

Nevertheless, in hypercholesterolemic patients, measurement of serum concentrations of circulating LOX-1 has been associated with the imbalanced regulation of L-arginine-NO pathway [16]. The consequent overproduction of reactive oxygen and nitrogen species (ROS and RNS, respectively) might be derived by the interaction between oxLDLs and LOX-1, leading to the activation of NADPH oxidase in vascular cells. The production of superoxide anions by NADPH oxidase, in turn, determines a reduced NO bioavailability, which leads to the dysregulation of the eNOS/iNOS balance and, finally, to endothelial dysfunction [17,18], thus promoting the atherosclerotic process [19]. A further effect of oxidative stress underlying endothelial dysfunction, is represented by the direct action of free radicals on eNOS. In fact, their overproduction causes eNOS uncoupling, which is unable to produce NO, but generates superoxide anions, thus amplifying the oxidative damage [20]. In this context, the reduced NO levels, as well as the increase of lipoperoxides, can upregulate iNOS activity to compensate for impaired NO bioavailability. Despite the activation of this adaptive mechanism, lipoperoxides exert a detrimental effect on endothelial cells in response to hypercholesterolemia. Indeed, their increase causes an enhanced expression of LOX-1 and, consequently, a further dysregulation of NO production [21].

Overall, these mechanisms represent the basis for the correlation between dysregulation of eNOS/iNOS enzymes induced by high levels of oxLDLs internalization, LOX-1 overexpression, and inflammation of vascular tissues, leading to atherosclerosis development and progression (Figure 1). In accordance with this hypothesis, it has been demonstrated that the increased expression of LOX-1 represents a consequence of increasing concentrations of interleukin 6 (IL-6) indicative of the development of inflammation in microvasculature [22].

The present review summarizes some recent advances in the bio-molecular mechanisms involved in the imbalanced modulation of NO biosynthesis and release, leading to EC dysfunction. In addition, the effect of oxLDL-dependent impairment of the NO-related pathway on proinflammatory mechanisms in the development of atherosclerosis are focused on.

### Mechanisms of *NO* Release and *NOS* Regulation

The generation of NO represents the product of the oxidation of the terminal guanidino nitrogen of L-arginine catalyzed by NOS through a highly-regulated process, which is mainly based on the utilization of the endogenous pool of L-arginine as a substrate. All isoforms of NOS utilize co-substrates, such as molecular oxygen, reduced nicotinamide-adenine-dinucleotide phosphate (NADPH), flavin adenine dinucleotide (FAD), flavin mononucleotide (FMN), and (6*R*-)5,6,7,8-tetrahydro-l-biopterin (BH_4_). The generation of NO from L-arginine by NOS isoenzymes occurs via formation of an intermediate, *N*^ω^-hydroxy-l-arginine, that is, in turn, oxidized to L-citrulline and NO in the late stage of the reaction. In the process of constitutive formation of NO, calmodulin is modulated by the increase of intracellular Ca^2+^ levels, thereby enhancing the flow of electrons from NADPH, which represents the signal for activating the reductase function of the enzyme. This explains the differences seen among constitutive and inducible NO-generating enzymes (Figure 2). Indeed, under basal conditions, the release of NO from eNOS is pulsed and tightly dependent on the rise of Ca^2+^ intracellular levels. This, in turn, leads to strong binding of calmodulin to the enzyme, which generates nM concentration of NO [23].

Apart from Ca^2+^-calmodulin-dependent mechanisms of eNOS regulation, other proteins have also been shown to contribute in the basal release of NO from ECs. In particular, evidence exists that heat shock protein 90 (hsp90) has been found to lead to allosteric activation of eNOS [24,25]. On the other hand, eNOS is inhibited by caveolin-1 protein, an effect counteracted by calmodulin and hsp90 [26].

eNOS utilizes co-substrates, such as NADPH, FAD, FMN and BH_4,_ to generate NO from L-arginine, which is oxidized to L-citrulline and NO in the late stage of the reaction. In the process of constitutive formation of NO, calmodulin is modulated by the increase of intracellular Ca^2+^ levels, thereby enhancing the flow of electrons from NADPH, which represents the signal for activating the reductase function of the enzyme.

Finally, mechanisms other than Ca^2+^ modulation may activate eNOS via serine (Ser 1177, in particular), threonine, as well as tyrosine phosphorylation of the enzyme. Indeed, bradikinin, the vascular endothelial growth factor (VEGF), and shear stress, which represent the physiological stimuli responsible for pulsed, basal release of NO, thereby contributing to small vessel regulation, have been shown to produce their effect via phosphorylation of Ser 1177 and increased Ca^2^+ sensitivity of eNOS [27]. On the other hand, Akt-1-mediated serine/threonine phosphorylation contributes in eNOS activation, mostly carried out by insulin release. In contrast, dephosphorylation of Thr495 seems to be associated with increased calmodulin-dependent activation of eNOS, though this mechanism is still to be better characterized.

The three major isoforms of NOS have been classified on the basis of several parameters; in particular, amino acid sequences (only 50%–60% identity), tissue and cellular localization, and different regulation mechanisms have been considered. The calcium/calmodulin-dependent NOS isoforms are expressed constitutively (cNOS) in endothelial cells (eNOS) and brain tissue (astrocytes and neurons in particular; nNOS), respectively [28,29,30]. The third NOS isoform is represented by the cytokine-inducible and calcium/calmodulin-independent form, named iNOS [28,29,30,31,32]. The distinct properties of each NOS isoform, such as the features of activation and the tissue localization, define both physiological or pathophysiological effects [33]. For example, the release of NO from cNOS occurs transiently, at nanomolar concentrations, and cNOS-derived NO regulates organ blood flow distribution in the cardiovascular and renal system. In particular, at the vascular level, it inhibits platelet aggregation, platelet and leukocyte adhesion, and smooth muscle cell proliferation, whereas it promotes diuresis and natriuresis within the kidney [28,29,34]. In addition, it is involved in neurotransmission [35,36]. These effects are mainly due to the binding of NO to Fe^2+^ in the heme prosthetic group of soluble guanylate cyclase, responsible for the conversion of GTP to cyclic GMP [28,29].

On the other hand, iNOS is overexpressed in response to different inflammatory stimuli, such as endogenous cytokines and bacterial lipopolysaccharide endotoxin (LPS), and causes a delayed, but persistent, synthesis of a large amount of NO [28,29,32].

The cross-modulation of eNOS and iNOS actions in cardiovascular systems represents a crucial event, aimed to ensure the right equilibrium between the constitutive, anti-atherogenic release of pulsed NO at nanomolar concentrations, and the suppression of its inducible release at micromolar concentrations [37]. Thus, the maintenance of this balance reduces the potentially dangerous effect of NO overproduction by iNOS, which, after the reaction with superoxide anions, can trigger peroxidative processes (Table 1) [24,25,28,38].

It has been demonstrated that, in different tissues expressing both cNOS and iNOS, as well as in cells expressing only inducible iNOS, both NO and NO donors can influence the production of either isoforms of these enzymes [6,39]. In particular, under basal conditions, cNOS-derived NO causes the inhibition of iNOS expression, suppressing NF-kB signalling [40,41]. On the other hand, extracellular stimuli, such as bacterial endotoxin and endogenous inflammatory cytokine, are able to induce NF-kB, thus triggering iNOS expression [42,43]

The NO-related inhibition of NF-κB is characterized by different mechanisms, which do not affect the activation and translocation of NF-kB and are characterized by the interference of the interaction between NO and DNA. In particular, NO is able to modify the binding process of NF-κB to its promoter response element [44,45] and it has been shown to also inhibit NF-κB/DNA binding through S-nitrosylation of the Cys 62 residue of p50 subunit [46].

In addition, NO interacts with NF-κB by stabilizing its endogenous inhibitor IkB [46]. Indeed, both NO and NO donors are able to prevent IkB degradation and its dissociation from NF-kB, and to increase IkB mRNA expression without modulating mRNA expression of NF-kB (p65 and p50 subunits) [47]. As a consequence, after activation of iNOS by extracellular inducers of NF-kB, the early inhibitory effect of NO is replaced by its overproduction, which represents the main cause of peroxynitrite formation [48]. Peroxynitrite, in turn, probably via nitration of IkK, favors the shift of NF-kB in its active state, thus allowing iNOS activation [46].

## 2. The Effect of oxLDLs in the Modulation of NOS Isoforms

Clear evidence suggests that oxLDLs produce early endothelial dysfunction. Indeed, the occurrence of high concentrations of circulating oxLDLs, as found in hyperlipemic patients as well as in subjects undergoing metabolic syndrome, has been associated with an altered reactive vasodilatation, which represents the early stage of endothelial dysfunction [49,50]. The mechanism underlying oxLDL-related endothelial dysfunction and inflammation of vascular tissues is still unclear. However, the relationship among the internalization of lipoproteins, the modulation of EC mediators, such as NO, and the activation of transcription factors, such as NF-kB, appears to have a relevant role in the development of atherosclerosis, though the mechanism needs further studies In particular, it has been hypothesized that oxLDLs may suppress constitutive NO release via direct or indirect inhibition of eNOS, thereby leading to exaggerated activation of iNOS and subsequent toxic effects due to the reaction of NO with superoxide anions, generating peroxynitrite, which is responsible for EC damage [51]. In this context, a crucial role is played by the activity of the caveolae/NO coupled system.

Caveolae, 50–100 nm vesicular invaginations of the cell plasma membrane, represents the site of the important mechanisms occurring at the plasma membrane, such as vesicular trafficking and signal transduction [52]. The scaffolding protein Caveolin-1 is the main structural and signalling component of caveolae in different cells, including ECs, which may interact with several molecules, thereby promoting important signalling functions [53].

eNOS is mostly targeted to caveolae in the plasma membrane of EC through interaction with Caveolin-1 [53]. In particular, the binding of caveolin-1 to eNOS is a negative regulator of eNOS activity, and the hypercholesterolemia-induced decrease of NO production is probably due to enhanced interaction of caveolin-1 with eNOS [54,55], suggesting its involvement in endothelial dysfunction.

The regulatory role of caveolae in endocytosis and transcytosis processes occurring in ECs is mainly influenced by its carrier function aimed to mediate the uptake and transcytosis of oxLDLs. This is confirmed by the action of two inhibitors, filipin and nocodazole, which counteract oxLDL uptake and transcytosis, attenuating the crosstalk between oxLDLs and caveolae across the endothelial cell membrane [56].

OxLDLs have been also shown to upregulate caveolin-1 time-dependently, thus promoting the translocation of NF-κB and regulating the transcription of iNOS and LOX-1 [56]. In this scenario, caveolin-1 might promote oxLDL uptake by ECs and NF-κB might be involved in this pathway.

The activity of Caveolin-1 is regulated by TRL4. In particular, evidence exists that Caveolin-1 Tyr14 phosphorylation is a crucial step in TLR4 signalling and, consequently, it mediates the inflammatory response in ECs [57]. In particular, TLR4-mediated recruitment of the adaptor protein MyD88 induces the phosphorylation and the degradation of IkB, favoring an early activation and translocation of NF-κB into the nucleus and, finally, cell death [57].

It has been hypothesized that high mobility group box 1 protein (HMGB1) may play a crucial role in the oxLDL-induced development of atherosclerosis [58]. Furthermore, on the basis of the studies on HMGB1, it has been suggested as a potential role for oxLDL-mediated TLR4/Caveolin-1 expression in ECs, which represents a negative modulator of eNOS activity and induces high permeability in the EC layer [59].

In summary, oxLDLs impair the balance between constitutive eNOS and inflammatory inducible iNOS in ECs. This occurs through enhanced expression of Caveolin-1, probably via HMGB1 and subsequent TLR4 signalling activation. Moreover, increased levels of TLR4 can induce the expression of LOX-1 because of the presence of a positive feedback between TLR-4 and LOX-1 [60]. In turn, LOX-1, which is also directly induced by oxLDLs, promotes the translocation of NF-kB into the nucleus. This inhibits protective mechanisms, such as eNOS function and protective autophagy, finally leading to EC apoptosis and subsequent endothelial dysfunction (Figure 3).

In particular, oxLDLs modulate HGMB1/TRL4/Caveolin-1 signalling in the eNOS/iNOS balance, which impairs autophagic/apoptotic responses in vascular and nonvascular cells.

## 3. Role of Scavenger Receptor LOX-1 in iNOS Modulation

Evidence exists that vascular inflammation is mediated by several mechanisms, which are mainly due to the overexpression of the lectin-like oxidized LDL (LOX-1), a scavenger receptor that selectively internalizes oxLDLs in ECs [61,62].

In particular, recent data suggest that an increased generation of oxLDLs leads to vascular inflammation via overexpression of LOX-1 receptor in EC, which, in turn, is accompanied by an imbalanced production of NO, attenuation of protective autophagy, and, in the late stages, by increased EC apoptosis. Indeed, evidence exists that silencing the LOX-1 receptor via ShRNA restores autophagy and protects against oxLDL-induced apoptotic cell death, thus suggesting the essential role of LOX-1 in mediating oxLDL-dependent impairment of autophagy [63].

Similar results have also recently been found in animal models of atherosclerotic disorders in carotid arteries. Indeed, it has been demonstrated that oxidized LDL uptake through LOX-1 contributes to induced endothelial dysfunction observed in the early stages of this pathology [59,64], though the pathophysiological mechanisms involved in oxLDL/LOX-1-related dysfunction of ECs remain to be elucidated.

LOX-1 is known to represent a type II transmembrane protein of 50 kDa, belonging to the C-type lectin family, and it has high affinity for oxLDLs, which are incorporated into ECs as a consequence of the endocytotic process. The LOX-1 receptor, as previously shown in bovine aortic ECs, contains 273 amino acid residues and is encoded by a single copy gene located in the p12.3–p13.2 region of human chromosome 12, indicated as lectin-like oxidized low-density lipoprotein receptor 1 (*OLR1*) gene. The structure of the LOX-1 receptor is characterized by a short N-terminal cytoplasmic domain and a long C-terminal extracellular domain [65].

The interaction of oxLDLs with the LOX-1 receptor, located diffusely at the EC surface, represents the major component for activating its expression in the vascular wall. However, recent studies have displayed that many factors, including the release of cytokines, shear stress, and vascular injury, may contribute in up-regulating LOX-1 [66,67,68]. Moreover, a LOX-1 overexpression is found under pathological conditions, such as hypertension, diabetes, and hypercholesterolemia [69,70,71,72,73]. The overexpression and subsequent activation of LOX-1 by oxLDLs is followed by an inflammatory response, which is mainly due to a prominent activation of adhesion molecules, such as VCAM-1 and ICAM-1, chemotactic factors, such as MCP-1, and, finally, to exaggerated free radical species formation and oxidative stress [74,75,76]. All these processes contribute to the enhanced endothelial adhesiveness of leukocytes and to an increase in chemokine expression.

In particular, an increased production of reactive oxygen species (ROS), such as superoxide anions (O_2_^-^), occurs directly via LOX-1-induced NADPH oxidase activation [77,78]. In this context, ROS overproduction and the subsequent LOX-1 activation seems to impair the PI-3-kinase/Akt pathway, causing an early attenuation of constitutive eNOS activity through inhibition of its phosphorylation/activation [78,79].

The central role of eNOS dysfunction in the onset of atherosclerosis has been further confirmed by in vitro discoveries showing that in ECs, oxLDLs, via LOX-1 activation, elicited a time-dependent decrease in serine 1179 phosphorylation of eNOS, causing its inactivation and, at the same time, the impairment of physiological NO-mediated suppression of iNOS gene expression [41,80]. As a consequence, NO production by iNOS causes the generation of high levels of peroxynitrite, which has been correlated with EC death via apoptosis, as demonstrated by the enhancement of caspase-3 expression [81,82].

The involvement of oxidative stress triggered by the oxLDLs/LOX-1 interaction was confirmed by detecting the effect of N-acetylcysteine (NAC), a thiol-containing radical scavenger and glutathione precursor, able to counteract EC mortality induced by oxidative damage [83]. Moreover, a similar effect was observed after recovering physiological NO levels, an effect achieved by treating Bovine aortic endothelial cells (BAEC) BAEC with the NO donor S-nitroso-N-acetylpenicillamine (SNAP) [63].

This direct correlation between LOX-1 activation and free radical-induced impairment of EC has been also proven by using pterostilbene, a naturally occurring analogue of antioxidant resveratrol, which has been shown to inhibit oxLDL-induced apoptosis of human ECs through the down-regulation of LOX-1 expression and the suppression of intracellular oxidative stress [81,84].

Inflammation represents a relevant aspect in the mechanisms involved in the reduction of NO bioavailability, induced by hypercholesterolemia via oxLDLs/LOX-1 interaction [11]. In fact, recent evidence suggests that inflammatory stimuli, such as the *Escherichia coli* Lipopolysaccharide (LPS), enhance LOX-1 expression via the TLR4 pathway, thus leading to enhanced formation of iNOS and, subsequently, to vascular damage. In particular, it has been clearly shown that, in human umbilical vein endothelial cells (HUVECs), treatment with LPS leads to LOX-1 mRNA and protein overexpression in a time- and concentration-dependent manner. The protein expression of LOX-1 was accompanied by the enhanced phosphorylation of p38MAPK and p65 both in vitro and in vivo. On the other hand, LPS-induced LOX-1 expression was blocked by siRNA for TLR4, MyD88, and Nox4, as well as by p38MAPK, NF-κB, cyclooxygenase-2, and Nox inhibitors. In addition, oxLDLs/LOX-1 interaction induced by LPS leads to endothelial-monocyte adhesion, which was prevented by anti-LOX-1 antibody. Therefore, LPS induces LOX-1 expression via the TLR4/MyD88/ROS activated p38MAPK/NF-κB pathway in endothelial cells, suggesting new regulatory mechanisms for LOX-1 expression.

These mechanisms have relevant consequences in the relationship between oxLDLs/LOX-1/pro-inflammatory responses and the imbalanced production of NO in the development of atherosclerotic process. Indeed, in microvascular endothelial cells, the expression of inflammatory cytokines, such as IL-6, increased in a dose-dependent manner in the presence of moderate to high levels of LDLs and oxLDLs. The incubation of the cells with IL-6 induced an evident up-regulation of LOX-1 and iNOS expression, as previously mentioned [22]. Other studies also indicated that the soluble IL-6 receptor (sIL-6R gp80), by binding IL-6, could induce LOX-1 and iNOS in ECs [85,86]. Therefore, the increase of oxLDLs and the presence of inflammation represent the main factors responsible for LOX-1 expression and, consequently, for the reduction of plasma oxLDLs. This evidence might explain the rapid reduction of cholesterol during the acute phase of inflammatory disease. In this context, despite a reduced plasma level of oxLDLs, a low release of constitutive NO by ECs might damage the endothelium, as demonstrated by recent discoveries [22].

Thus, alterations in NO regulation due to oxLDLs-induced LOX-1 overexpression at the level of microvessels might be early predictors of atherosclerosis and of an adverse cardiac outcome [87]. Moreover, the oxLDLs/LOX-1 interaction and the imbalanced modulation of the cNOS/iNOS relationship also appears to play a key role in vascular dysfunction because it represents the cause of the alteration of matrix metalloproteinase (MMP) production and activation, which causes atherosclerotic plaque rupture or erosion [88,89,90].

## 4. The Role of LOX-1/iNOS Activation in the Crosstalk between Apoptosis and Autophagy of ECs

In the few last years, it has been hypothesized that a more complex mechanism responsible for EC death via apoptosis exists. In particular, Lu et al. showed that LOX-1 activation by L5, an electronegative component of LDL abundant in dyslipidemic, but not in normolipidemic, human plasma, selectively inhibited Bcl-xL and anti-apoptotic Bcl-2 expression, and Akt and eNOS phosphorylation. Moreover, LOX-1 activation induced an enhancement in pro-apoptotic Bax and Bad expression. Finally, L5 has been shown to cause the activation of caspase-3 and mitochondrial release of cytochrome c, thus further favoring apoptosis [82,91,92].

The Bcl-2 protein family is divided into two different subgroups based on the presence of their Bcl-2 homology (BH) domain(s): anti-apoptotic proteins, such as Bcl-2 and Bcl-xL, and pro-apoptotic proteins, such as Bad and Bax. Bcl-2 family proteins regulate autophagy, which is also considered to be a cellular defensive mechanism able to eliminate ROS-induced damaged proteins [93,94].

Beclin 1 is an important effector of autophagy, belonging to the Bcl-2 family proteins, which binds to a hydrophobic groove in Bcl-2/Bcl-xL, similarly to pro-apoptotic proteins of Bcl-2 family. The formation of this complex, in turn, impairs autophagy and its inhibitory effect can be suppressed by the dissociation of Beclin 1 mediated by pro-apoptotic proteins [95,96]. This suggests that apoptosis and autophagy may be co-regulated in the same directions. As a consequence, in the presence of stimuli (e.g., LOX-1 activation by L5 or oxLDLs) able to down-regulate Bcl-2/Bcl-xL and to induce pro-apoptotic proteins [97,98], it is conceivable that the switch between autophagy and apoptosis is regulated through alternative mechanisms.

In particular, it has been demonstrated that Beclin 1, being a direct caspase substrate, can lose its autophagy-inducing property after its cleavage. Indeed, after the direct interaction of its C-terminal fragment with mitochondria, it causes the release of pro-apoptotic factors, thus promoting apoptosis [99].

In vitro experiments have supported this theory. In particular, under starvation, an experimental setting able to activate autophagic response, a reduced expression of oxLDLs-induced Beclin 1 was observed, and this effect also correlated to an increased expression of iNOS and caspase-3. This suggests that the oxidative stimulus was not able to restore physiologic NO levels early, favoring free radical overproduction and caspase-induced Beclin-1 cleavage. The results of these mechanisms triggered by oxLDLs was represented by a suppression of autophagy associated with an enhanced apoptosis (Figure 4) [63,100].

The balance between protective autophagy and apoptotic cell death by LOX-1 has also been further confirmed by the up-regulation of another marker of autophagy, LC3 II (microtubule-associated protein 1A/1B-light chain 3 type II). Indeed, oxLDLs led to a downregulation of LC3 II marker, whereasLOX-1 silencing, induces an up-regulation of LC3 expression [63,100,101],

LOX-1 activation by L5 inhibits Bcl-xL, Bcl-2 and Akt expression. This induces an enhancement in pro-apoptotic Bax and Bad expression, cytochrome C release, and the activation of caspase-3 favoring apoptosis.

The co-regulation between apoptosis and autophagy is mediated by the Beclin 1-Bcl2 complex. LOX-1 activation by L5 promotes the down regulation of Bcl-2, thus promoting apoptosis and, on the other hand, Beclin-1 dissociation. In turn, free Beclin-1 is cleaved by caspases blocking autophagic process.

## 5. Conclusions

Taken together, the data reported in this review article suggest that oxLDLs lead to the direct and indirect impairment of constitutive release of NO, highlighting the overexpression of inducible iNOS. This effect, combined with overproduction of ROS, leads to endothelial cell dysfunction and subsequent development of atherosclerosis. The activation of LOX-1 receptor and subsequent activation of NFkB represent key events in this complex dysregulation of NOS isoforms, contributing to the attenuation of protective autophagic response and an accelerated EC apoptotic death, which is the end stage of endothelial dysfunction. This information allows us to better define the pathophysiology of imbalanced regulation of NOS isoforms, which occurs at early stages of the atherosclerotic process and represents a perspective for selective therapeutic interventions in cardiovascular disease prevention.

## Figures and Tables

**Figure 1 ijms-20-03294-f001:**
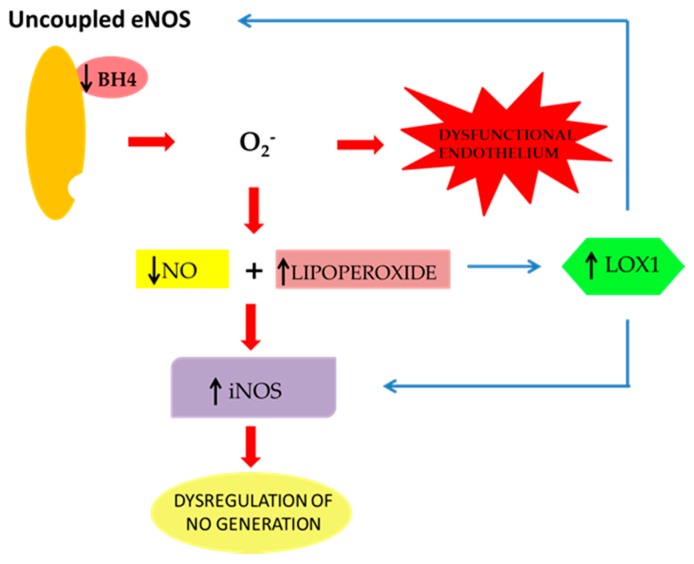
Basis for the correlation between dysregulation of eNOS/iNOS enzymes. Uncoupled eNOS is unable to produce NO and generates superoxide anions. The reduction in NO levels and the increase in lipoperoxides upregulate iNOS activity. In this context, lipoperoxide increase causes an enhanced expression of LOX-1 and, consequently, a further dysregulation of NO production.

**Figure 2 ijms-20-03294-f002:**
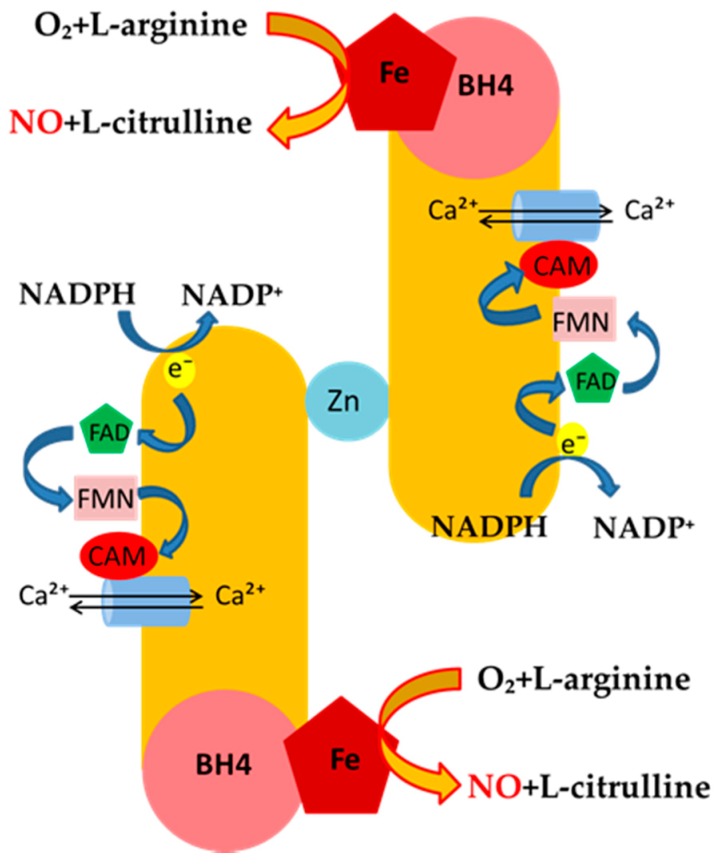
Mechanisms of NO-release and eNOS regulation.

**Figure 3 ijms-20-03294-f003:**
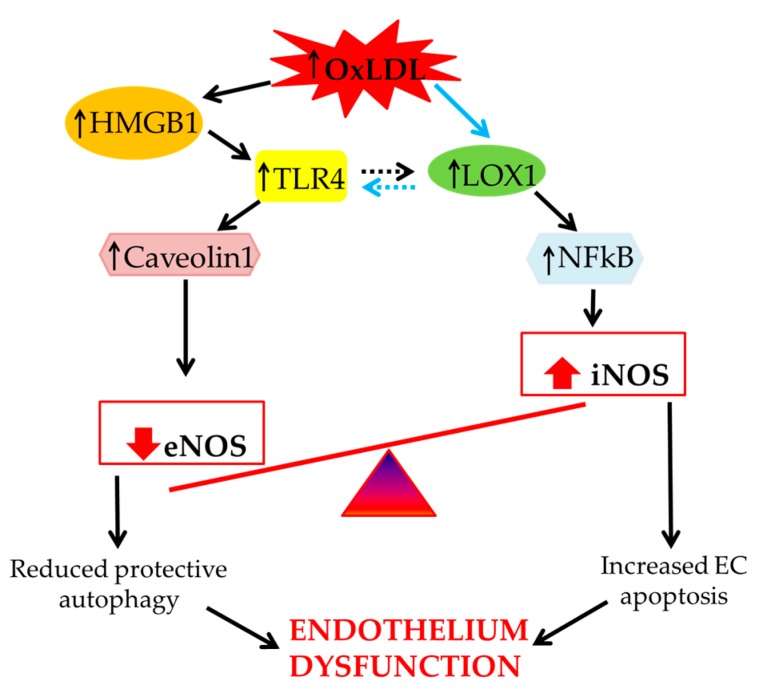
Proposed mechanism of oxLDL–related endothelial dysfunction via imbalanced regulation of the eNOS/iNOS relationship.

**Figure 4 ijms-20-03294-f004:**
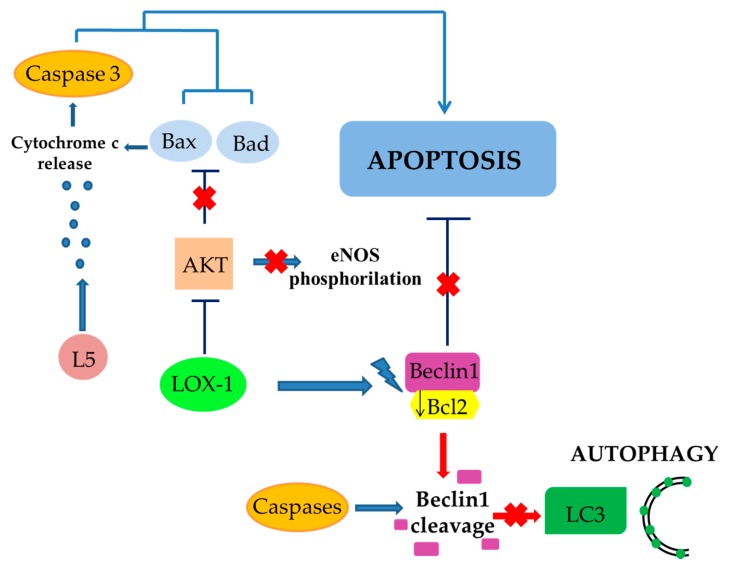
Role of LOX-1 activation in the crosstalk between apoptosis and autophagy in endothelial cells.

**Table 1 ijms-20-03294-t001:** Expression and functions of different nitric oxide synthase (NOS) isoforms.

NOS Isoform	Expression	Function
eNOS	endothelial	Under basal conditions, the release of NO from eNOS is pulsed and tightly dependent on the rise of Ca^2+^ intracellular levels. This, in turn, leads to strong binding of calmodulin to the enzyme, which generates nM concentration of NO, which regulates: organ blood flow distribution in the cardiovascular and renal system; inhibition of platelet aggregation, platelet and leukocyte adhesion and smooth muscle cell proliferation at vascular level; and promotion of diuresis and natriuresis within the kidney [24,25,30].
nNOS	neuronal	It is involved in neurotransmission in astrocytes and neurons [31,32].
iNOS	inducible	It is overexpressed in response to different inflammatory stimuli, such as endogenous cytokines and bacterial lipopolysaccharide endotoxin (LPS) and causes a delayed, but persistent, synthesis of a large amount of NO. Leads to endothelial dysfunction and, in the late stages, to the development of atherothrombosis. NO production by iNOS causes the generation of high levels of peroxynitrite, which has been correlated with endothelial cell (EC) death via apoptosis [24,25,28].

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
