# Peer review of "Modulation of Nitric Oxide Synthases by Oxidized LDLs: Role in Vascular Inflammation and Atherosclerosis Development"

_ijms, 2019, doi:10.3390/ijms20133294_

Round 1

Reviewer 1 Report

In the present manuscript, the authors summarized the importance of oxyLDL in endothelial dysfunction, vascular inflammation and atherosclerosis development. The current review mainly focused on how (mechanisms) oxyLDL produce impairment eNOS/iNOS machinery and play an role in endothelial dysfunction and pathogenesis of atherosclerosis.

Overall, the authors have done very good work in summarizing current understanding of the oxyLDL role in modulating eNOS/iNOS machinery in vascular inflammation and atherosclerosis development.

Here are my suggestions:

1)      I do see the authors mentioned only one overall summary figure in the manuscript. However, different sections of the manuscript have provided a lot of information and mechanisms and sometime difficult to follow readers. I would suggest including more figures/tables at different sections of the manuscript. It will help readers to follow and understand.

2)      Extensive English language editing is required throughout the manuscript and try to avoid more complex sentences.

3)      Page 2: last four paragraphs: It is very difficult to follow the correlation between dysregulated eNOS/iNOS and oxyLDL in these four paragraphs. I would suggest authors to revise them to make the concept very clear.

Author Response

Reviewer #1: 1) I do see the authors mentioned only one overall summary figure in the manuscript. However, different sections of the manuscript have provided a lot of information and mechanisms and sometime difficult to follow readers. I would suggest including more figures/tables at different sections of the manuscript. It will help readers to follow and understand.

R1: We thank the reviewer and we agree with his comments. More figures and tables at different sections of the manuscript were added, as suggested.

Reviewer #1: 2) Extensive English language editing is required throughout the manuscript and try to avoid more complex sentences.

R2: English has been revised, moreover complex sentences have been shortened as suggested.

Reviewer #1: 3) Page 2: last four paragraphs: It is very difficult to follow the correlation between dysregulated eNOS/iNOS and oxyLDL in these four paragraphs. I would suggest authors to revise them to make the concept very clear.

R3: We revised the paragraphs as required. We believed that now, this part is more clear.

Reviewer 2 Report

Date: June 19, 2019

Manuscript: Review

Title:Modulation of nitric oxide synthases by oxyLDL: role in vascular inflammation and atherosclerosis development

Authors: Micaela Gliozzi et al. 

This is a comprehensive review focuses on the effect of oxidized LDLs on endothelial dysfunction and atherosclerosis viaHMGB1, TLR4 and LOX-1.

Critics:

1.    In the Abstract, the first paragraph is very confusing.  Does authors mean the quantity of NO affect the development of atherosclerosis. If so, how much is the threshold? Otherwise, the authors need to rephrase the sentence. 

2.    In the introduction, Lane 59, the reference 6 is a review on Lox-1, which does not correspond to the comment.

3.    The oxidized LDLs in the scientific field uses abbreviation as oxLDLs.  In this review, the authors use “oxyLDL”.  It might create unnecessary confusing in terminology.

4.    In the introduction, Lane 74 to 80, the whole paragraph was very confusing.  Please rewrite to be clearer.

5.    Lane 81 to 85, needs a Reference for the comment.

6.    Lane 86, what is “poorly active NO”?

7.    Ref-16 on lane 89 does not correspond to the statement.

8.    Page 4, lane 147-152.  It is hard to understand that overproduction of NO had such a detrimental effect on endothelial cells.  Are there other publications support the statements?

9.    It is still unclear to page 5 whether increased of oxLDL downregulated the HMGB1/TLR4/Cavelin-1 to reduce eNOS or increased oxLDL binds to Lox-1 to induce NFkB, then iNOS. I wished the authors would be able to clear state this in this review.

10.  Does TLR4 interact with LOX-1?

Author Response

Reviewer #2: 1) In the Abstract, the first paragraph is very confusing.  Does authors mean the quantity of NO affect the development of atherosclerosis. If so, how much is the threshold? Otherwise, the authors need to rephrase the sentence.

R2:We thank the reviewer for the comment. We preferred to rephrase the sentence and make the abstract clearer.

Reviewer #2: 2) In the introduction, Lane 59, the reference 6 is a review on Lox-1, which does not correspond to the comment.

R3: We thank the reviewer for the elucidation. We updated the reference, which is now “Ulrich Förstermann, William C. Sessa. Nitric oxide synthases: regulation and function. Eur Heart J. 2012, 33: 829–837.” The reference is now the number 7.

Reviewer #2: 3) The oxidized LDLs in the scientific field uses abbreviation as oxLDLs.  In this review, the authors use “oxyLDL”.  It might create unnecessary confusing in terminology.

R3: Following the reviewer’s comment the confusing abbreviation “oxyLDL” has been replaced. In the text, now the abbreviation is “oxLDLs”, as suggested.

Reviewer #2: 4) In the introduction, Lane 74 to 80, the whole paragraph was very confusing.  Please rewrite to be clearer.

R4: The whole paragraph has been rewrite as suggested. We believe that we make it clear now.

Reviewer #2:  5) Lane 81 to 85, needs a Reference for the comment.

R5: The reference has been added. It is the ref number 19 in the text.

Reviewer #2:  6) Lane 86, what is “poorly active NO”?

R6: This part has been rephrased in “In this context, the reduced NO levels as well as the increase of lipoperoxides can up regulate iNOS activity to compensate impaired NO bioavailability.

Reviewer #2:7) Ref-16 on lane 89 does not correspond to the statement.

R6: We thank the reviewer for the observation. Ref. 16 has been changed. Please refer to ref.21

Reviewer #2: 8) Page 4, lane 147-152.  It is hard to understand that overproduction of NO had such a detrimental effect on endothelial cells.  Are there other publications support the statements?

R8: We thank the reviewer for the observation. NO overproduction by iNOS, which, after the reaction with superoxide anions, triggers peroxidative processes is supported by several evidences. Please see Refs 38, 24, 25, 28 and table 1.

Reviewer #2: 9) It is still unclear to page 5 whether increased of oxLDL downregulated the HMGB1/TLR4/Cavelin-1 to reduce eNOS or increased oxLDL binds to Lox-1 to induce NFkB, then iNOS. I wished the authors would be able to clear state this in this review.

R9: Thanks for the observation. On the basis of the studies on HMGB1, it has been suggested a potential role for oxLDL-mediated TLR4/Caveolin-1 expression in EC, which represents a negative modulator of eNOS activity and induces high permeability in EC layer.

Summarizing, oxLDLs impair the balance between constitutive eNOS and inflammatory inducible iNOS in EC. This occurs via direct modulation of caveolin-1 (probably via HMGB-1 and subsequent TLR4 signalling), which promotes the translocation of NF-kB into the nucleus. This inhibits protective mechanisms such as eNOS function and protective autophagy leading, at the end stage, to EC apoptosis and subsequent endothelial dysfunction. We stated it in the manuscript. 

Reviewer #2: 9. Does TLR4 interact with LOX-1?

R9: We thank the reviewer for giving us the opportunity to explain this concept. It is not clear how/whether TLR4 interact with LOX-1, but a very interesting paper showed that TLR4 knockdown, resulted in reduced LOX-1 expression and autophagy. This indicates a positive feedback between LOX-1 and TLR4. Moreover, also LOX-1 knockdown results in reduced expression of TLR4. The proposed mechanism is pictured in fig.3, ref. 60.